# Characterization of *Bacillus velezensis* EV17 and K-3618 and Their Polyketide Antibiotic Oxydifficidin, an Inhibitor of Prokaryotic Translation with Low Cytotoxicity

**DOI:** 10.3390/ijms262411777

**Published:** 2025-12-05

**Authors:** Alisa P. Chernyshova, Valeriya I. Marina, Andrey G. Tereshchenkov, Vladislava E. Sagitova, Maksim A. Kryakvin, Nikolai D. Dagaev, Eugeniya G. Yurchenko, Kseniya A. Arzamazova, Elena B. Guglya, Olga A. Belozerova, Sergey I. Kovalchuk, Margarita N. Baranova, Arsen M. Kudzhaev, Anton E. Shikov, Maria N. Romanenko, Alexander Yu. Rudenko, Vladimir K. Chebotar, Maria S. Gancheva, Maria E. Baganova, Mikhail V. Biryukov, Tatiana V. Panova, Maria G. Khrenova, Vadim N. Tashlitsky, Natalia V. Sumbatyan, Yulia V. Zakalyukina, Kirill S. Antonets, Anton A. Nizhnikov, Vladimir I. Polshakov, Stanislav S. Terekhov, Maria I. Zvereva, Olga A. Dontsova, Petr V. Sergiev, Vera A. Alferova, Dmitrii A. Lukianov

**Affiliations:** 1Center for Molecular and Cellular Biology, 121205 Moscow, Russia; abeidy@mail.ru (A.P.C.); juline@mail.ru (Y.V.Z.);; 2Department of Chemistry, Lomonosov Moscow State University, 199992 Moscow, Russiazverevame@my.msu.ru (M.I.Z.); 3Shemyakin-Ovchinnikov Institute of Bioorganic Chemistry, Miklukho-Maklaya 16/10, 117997 Moscow, Russiaxerx222@gmail.com (S.I.K.); alferovava@gmail.com (V.A.A.); 4A.N. Belozersky Institute of Physico-Chemical Biology, Lomonosov Moscow State University, 1/40 Leninskie Gory, 119992 Moscow, Russia; 5Faculty of Bioengineering and Bioinformatics, Lomonosov Moscow State University, 199992 Moscow, Russia; 6North Caucasian Federal Scientific Center of Horticulture, Viticulture, Winemaking, 350901 Krasnodar, Russiaarzamazova.99@mail.ru (K.A.A.); 7Biological Faculty, St. Petersburg State University, 199034 St. Petersburg, Russia; a.shikov@arriam.ru (A.E.S.); m.romanenko@arriam.ru (M.N.R.); a.nizhnikov@arriam.ru (A.A.N.); 8All-Russia Research Institute for Agricultural Microbiology, 196608 St. Petersburg, Russia; vladchebotar@arriam.ru (V.K.C.);; 9Department of Biology, Lomonosov Moscow State University, 119991 Moscow, Russia; metrim@gmail.com; 10Scientific Center for Translational Medicine, Sirius University of Science and Technology, 354340 Sochi, Russia; 11Bach Institute of Biochemistry, Federal Research Centre “Fundamentals of Biotechnology” of the Russian Academy of Sciences, 119071 Moscow, Russia; 12Soil Science Faculty, Lomonosov Moscow State University, 119991 Moscow, Russia

**Keywords:** oxydifficidin, antibiotics, antimicrobial activity, translation, *Bacillus velezensis*

## Abstract

Oxydifficidin is a natural polyketide antibiotic that has long been recognized as a ribosome-targeting agent that inhibits protein synthesis. In this paper, we describe *Bacillus velezensis* strain EV17 and compare its complete genome sequence with that of the previously characterized *B. velezensis* strain K-3618 and the difficidin biosynthetic gene cluster (BGC) combined with mass spectrometry to elucidate the production of oxydifficidin by these strains. Toeprinting and small fluorescent peptide assays showed that isolated oxydifficidin induces a generalized inhibition of translation at every step of elongation in protein biosynthesis. In previous studies, it has been demonstrated that oxydifficidin targets bL12 protein. Although spontaneous mutations conferring resistance to oxydifficidin in ribosomal protein bL12 located relatively close to the thiostrepton binding site on uL11, our data show that oxydifficidin binding does not interfere with thiostrepton, thereby refining previous findings about its putative ribosomal target. We are the first to show that this compound does not affect eukaryotic translation and has two orders of magnitude lower effect on eukaryotic cells compared to bacteria. These facts are important to further investigate its potential as a bioprotectant against phytopathogens or even as a therapeutic agent.

## 1. Introduction

The global rise in antibiotic resistance presents a critical threat to public health, necessitating the discovery and development of novel antimicrobial agents [1]. In the search for novel antimicrobials, the scientific community has traditionally focused on the discovery of entirely new chemical scaffolds. However, this approach is increasingly constrained by the limited success rate of high-throughput screens and the rediscovery of known compounds [2]. In contrast, an often-overlooked strategy involves the systematic re-evaluation of previously discovered antibiotics that were either shelved due to suboptimal pharmacological profiles, narrow spectra of activity, or incomplete understanding of their mechanisms of action [3]. With modern analytical tools—such as high-resolution mass spectrometry, various methods for assessing the mechanism of action, and genome-wide fitness profiling—it is now possible to reassess these molecules with far greater precision and depth [4]. Re-developing revived antibiotics involves systematically identifying knowledge gaps, conducting updated non-clinical and clinical studies, and applying modern scientific methods such as toxicity assessment and resistance monitoring [5,6].

Bacteria represent a major source of novel antibiotic compounds from natural environments. Among them, filamentous actinomycetes produce up to 64% of identified classes of natural antibiotics, and the rest are found in other bacteria and fungi [7,8,9]. Nevertheless, Gram-positive bacteria belonging to Firmicutes, in particular, *Bacillaceae* species, are also considered as important producers of structurally diverse classes of natural antibiotics including antibiotic polypeptides [10,11,12], polyketides [13,14], and lipopeptides [15,16,17,18].

Polyketides are structurally diverse natural metabolites exhibiting a wide range of biological activities, notably antimicrobial effects [15,19,20]. Many polyketide antibiotics are produced by soil bacteria, including *Bacillaceae* species, and contribute to the biocontrol of plant pathogens and potentially have agricultural application [13,15]. However, some polyketide antibiotics, such as oxydifficidin, were historically overlooked due to their low stability. Nevertheless, they remain valuable for providing insights into antibacterial mechanisms of action, motivating renewed investigation.

This work presents a multifaceted characterization of the natural polyketide antibiotic oxydifficidin (Oxy) and its producer strains. While Oxy is known to inhibit protein synthesis, its molecular mechanism remains understudied. A variety of molecular biology approaches were employed to address this issue. Specifically, toeprinting assays are used to analyze its impact on distinct stages of translation and refine the hypothesis, based on the previously published study [21], that it impedes the elongation cycle. At the same time, we report the genomic characterisation of strains EV17 and K-3618. Both strains are definitively assigned to *Bacillus velezensis* through whole-genome sequencing, phylogenetic analysis, digital DNA-DNA hybridization (dDDH) and average nucleotide identity (ANI) comparisons, demonstrating their taxonomic position and evolutionary kinship with renowned biocontrol lineages.

## 2. Results and Discussion

### 2.1. Classification of Strains EV17 and K-3618

#### 2.1.1. Genome Sequencing and Annotation

We have sequenced and assembled the genome of the oxydifficidin-producing strain EV17. The mean coverage of Illumina reads was 644. The assembly of the strain resulted in one circular contig—a full chromosome (1 contig), with a total genome length (equal to N50) of 3,978,750 base pairs (bp). The assembly exhibited a GC-content of 46.53%. CheckM [22] analysis of assembly quality revealed a nearly complete genome (98.82%) with no detectable contamination (0%).

Genome sequencing and annotation of the strain K-3618 were reported in our recent study [23]. Comparative testing of our *B. velezensis* strains EV17 and K-3618 was conducted against previously published strains FZB42 and NRRL B-41580^T^. According to our comparison of genes *amyE* and clusters of several antibiotics (Bacillaene, Fengycin, Macrolactin, Difficidin), we can see that the identity of the strains is 96% or higher, which proves that they are closely related (Table 1 and Appendix A) [24].

Overall, genome sequencing of endophytic and rhizospheric strains EV17 and K-3618 revealed that they belong to the species *B. velezensis*, a species often closely associated with plants, and well known for promoting plant growth and biocontrol [27]. Representatives of *B. velezensis* are multifunctional Plant Growth-Promoting Rhizobacteria (PGPR) [15,27,28,29,30]. They act as biofertilizers, biopesticides, and inducers of stress tolerance in economically important agricultural crops. These bacteria enhance plant development by synthesizing phytohormones (auxins, gibberellins), solubilizing essential nutrients (phosphorus, potassium), and producing siderophores such as bacillibactin improve iron uptake. They provide robust protection against fungal and bacterial pathogens by producing a diverse arsenal of antimicrobials (lipopeptides, polyketides, etc.), secreting cell wall-degrading enzymes (chitinases, glucanases), and triggering Induced Systemic Resistance (ISR), which primes the plant’s own immune defenses [28]. Since *B. velezensis* strains are habitual inhabitants of the rhizosphere, rhizoplane, and plant endosphere, biocontrol agents based on *Bacillus* spores practically do not disrupt the composition of the natural microbial communities of plant roots [15].

Due to their ability to inhibit a wide range of phytopathogens (bacteria, fungi, and nematodes), promote plant growth, and produce spores and biofilms that simplify storage and use [28,31,32] these bacteria are commonly used as the foundation for well-known commercial bioproducts (e.g., RhizoVital^®^, Amylo-X^®^ WG, and Taegro^®^). Strains EV17 and K-3618 are the most closely related to the commercially used *B. velezensis* FZB24 (Taegro^®^), previously known as the type-strain of *B. amyloliquefaciens* subsp. *plantarum* [25].

#### 2.1.2. Whole Genome Phylogeny

The phylogenetic analysis based on whole-genome sequences showed that both strains EV17 and K-3618 formed a moderately supported clade with strains belonging to *B. velezensis* with 81% bootstrap value (Figure 1): in addition to the type representative of *B. velezensis*, NRRL B-41580^T^, described in 2005 [26], this clade includes strains KACC 13105, originally described as *Bacillus methylotrophicus* sp. nov. [33], strain FZB42 known since 2011 as *Bacillus amyloliquefaciens* subsp. *plantarum* [25], but later reclassified as *B. velezensis*, and two strains, “*Bacillus oryzicola*” and “*Bacillus ayatagriensis*”, which now have unconfirmed nomenclatural status.

To confirm the taxonomic identification, we calculated average nucleotide identity (ANI) between EV17 and K-3618 and closest genomes from *B. velezensis* clade. This revealed that EV17 and K-3618 share more than 98.0% ANI with the genomes that are currently classified as *B. velezensis* (Table 1).

Digital DNA-DNA hybridizations (DDH) indicated that DNA–DNA relatedness between EV17 and K-3618, from one hand, and type strain *B. velezensis* NRRL B-41580^T^, from other, are 85.8% (Table 2), which is more than the cut-off point of 70% recommended for the assignment of bacteria strains to the same genomic species [36].

Therefore, the ANI and dDDH values supported the conclusion that strains EV17 and K-3618 should be considered as representatives of *B. velezensis*.

### 2.2. Oxydifficidin Isolation

*B. velezensis* strains coded as EV17 and K-3618 were cultivated for antibacterial metabolites production and tested on *E. coli* Δ*tolC* pDualRep2 and *E. coli lptD^mut^* pDualRep2.1. *E. coli* Δ*tolC* pDualRep2 system lacks *tolC* gene, which encodes the TolC outer membrane channel—a key component of the multidrug efflux pumps in Gram-negative bacteria. The absence of TolC prevents the active efflux of toxic compounds, making the strain highly susceptible to antibiotics.

*E. coli lptD^mut^* pDualRep2.1 lacks 23 amino acids in LptD protein, which causes disruption in lipopolysaccharide assembly in the outer membrane, leading to increased permeability and heightened sensitivity to antibiotics. These reporter strains expressed fluorescent proteins in response to sublethal antibiotic concentrations, depending on their mechanism of action. Activation of the SOS response induced TurboRFP expression, while translation inhibition, characterized by ribosome stalling on mRNA, triggered Katushka2S expression [38].

Since broth culture demonstrated significant antibacterial activity against *E. coli* Δ*tolC* pDualRep2 (Appendix A), we carried out primary purification on LPS-500-H sorbent. Elution with 50–75% aqueous acetonitrile demonstrated strong induction of the reporter protein Katushka2S (Appendix A), similar to that observed with erythromycin (Ery), indicating that antibacterial compound produced by strains EV17 and K-3618 may negatively affect protein synthesis in bacterial cells. The most active fraction (eluted at about 75% ACN) was further analyzed by HPLC (Appendix A). HPLC fractions were tested for activity, and the fraction containing pure active substance was isolated and then analyzed by LC-MS.

### 2.3. Identification of Antibacterial Compound Oxydifficidin Produced by Strains EV17 and K-3618

Taking into account the plausible inhibition of protein synthesis by the active component, as well as the presence of genes demonstrating high similarity to difficidin BGC (BGC0000176), detected by the AntiSMASH in the whole-genome of EV17 and K-3618 (Figure 2A), we identified active compound by HR-LCMS. Analysis of the active fraction using HR-LCMS revealed an ion with *m*/*z* 583.2834, corresponding to the [M+Na]^+^ adduct of oxydifficidin (C_31_H_45_O_7_P; calculated mass 560.2936, Δ5.7 ppm), which is an oxidized form of difficidin [39], known to be produced by organisms, encoding the *dif* cluster [40,41]. In the same spectrum, characteristic fragment ions were observed at *m*/*z* 480.34 (M–phosphate), *m*/*z* 463.32 (M–phosphate–H_2_O) and *m*/*z* 445.31 (M–phosphate–2H_2_O), supporting the structural assignment (Figure 2B). Complementary HR-LCMS in negative mode revealed an ion with *m*/*z* 559.2698, which also confirmed the identification of oxydifficidin. Additional comparison of MS data (Appendix A), UV spectra (Appendix A) and the proposed mechanism of action against bacterial translation with previously published data [39] corroborated the production of oxydifficidin by the studied strains.

The 1H NMR spectrum of the isolated oxydifficidin sample (Appendix A) was in ex-cellent agreement with the previously reported data for this compound [21].

### 2.4. Biological Activity of Oxydifficidin

#### 2.4.1. Oxydifficidin Exhibits Antibacterial Activity

After isolation and purification of the active compound—oxydifficidin—the agar plate test was repeated (Figure 3). In both cases, strong reporter induction–evidenced by fluorescence—was observed, similar to that observed with Ery, suggesting that Oxy may have negative effect on protein biosynthesis in bacterial cells, as previously shown [39].

For subsequent experiments, we determined the minimum inhibitory concentration (MIC) of Oxy. An overnight culture of the *E. coli* BW25113 *lptD^mut^* strain [43] was diluted to an optical density (OD600) of 0.05–0.1, and oxydifficidin, along with the control antibiotic Ery, was added using a standard two-fold serial dilution method. Cultures were incubated overnight at 37 °C with shaking. MIC, defined as the lowest antibiotic concentration that completely inhibited visible bacterial growth, was determined to be 2.5 µg/mL for oxydifficidin against the *E. coli* BW25113 *lptD^mut^* strain.

#### 2.4.2. Oxydifficidin Inhibits Prokaryotic In Vitro Translation

To directly evaluate the ability of Oxy to inhibit protein synthesis, we employed in vitro translation system: a reconstituted system of purified components with reporter system based on luciferase mRNA. Translation efficiency was monitored by measuring luciferase activity, which depends on the enzymatic conversion of d-luciferin to oxyluciferin. Oxy was titrated over a concentration range spanning sub-MIC to supra-MIC levels as was previously established for *E. coli* BW25113 *lptD^mut^* strain (Figure 4B), with complete inhibition of protein synthesis observed at 2 µg/mL (3.6 µM). These results confirm that Oxy is a potent inhibitor of bacterial protein synthesis.

Previous publications have assessed in vitro inhibition of translation by difficidin. They have shown that 2 µg/mL (3.6 µM) completely inhibits the incorporation of radioactive amino acids in nascent peptides [39]. These data align with our findings for the difficidin derivative oxydifficidin, which likewise fully inhibits translation at 3.6 μM.

According to our research, Oxy’s main impact is on a general slowdown of protein biosynthesis, not a complete stop at a specific codon. In this context, the assumption that 3.6 µM is the definitive MIC for protein synthesis is an overestimation. The inhibition observed in the assay is due to the failure to complete long polypeptides, not due to targeted blockage at a defined location.

#### 2.4.3. Oxydifficidin Cause Generalized Inhibition of Translation

To investigate the mechanism of action of Oxy during translation, we performed both fluorescent and radioactively labeled toeprinting assays with fluorescent toeprint shown for the *ermCL* mRNA (Figure 4A) and radioactive toeprints on related template *ermDL* (Appendix A), to monitor ribosome positioning using reverse transcription. Reactions were conducted both in the absence of antibiotic and in the presence of Oxy, as well as the control antibiotics thiostrepton (Ths) and borrelidin (Borr). Ths served as a reference inhibitor of translation initiation, while Borr was used as a control for threonine-specific stalling, with the corresponding codon located in the second part of the transcript.

Inclusion of Borr in Oxy-treated samples allowed us to assess the extent of residual translation reaching the threonine codon. Only at the highest Oxy concentration (36 µM), the Borr-correlated stop was not detectable, indicating strong inhibition of translation during earlier rounds of elongation. At 3.6 and 0.9 µM Oxy, residual translation to the threonine codon was still observed, with the signal at 0.9 µM appearing broader than that at 3.6 µM, indicating a concentration-dependent manner of inhibition. Oxy did not produce distinct stalling sites but instead caused widespread inhibition, with stops distributed across multiple codons. This pattern suggests that Oxy lacks detectable context specificity and is likely capable of undergoing multiple rounds of binding and dissociation from the ribosome during elongation. In contrast, Ths treatment resulted in the distinct stop exclusively at the first codon, consistent with inhibition of initiation [44].

#### 2.4.4. Oxydifficidin Practically Does Not Inhibit Initiation Step of Bacterial Translation

To clarify the mechanism of action of oxydifficidin in more detail, an experiment was conducted using the MF2 template encoding a short peptide consisting of three amino acids—methionine and two phenylalanines [45]. Briefly, each translation product is labeled by the fluorescent N-terminal BODIPY (BPy) group introduced via initiator BPy-Met-tRNA^fMet^. This system allows visualization of disruptions at various stages of translation due to the unusual mobility of BODIPY-labeled products in RNA-urea PAGE.

We have shown that at high concentrations (36 μM), oxydifficidin completely suppresses the synthesis of the full-length short peptide, although slight band of dipeptide product is seen (Figure 4D). For positive control—madumicin (50 μM)—we do not visualize MF2 product due to the fact that madumycin blocks the formation of the first peptide bond. Not all antibiotics are capable of blocking the biosynthesis of BODIPY-MF2, so Ery was chosen as a negative control antibiotic, which does not inhibit protein translation due to the absence of the +X+ motif. Therefore, we can assume that oxydifficidin affects peptidyl-transferase reaction or translocation step of translation.

According to the toeprinting and BPy assays, oxydifficidin may also partially affect initiation. To directly test whether translation initiation is impacted, we employed a template encoding only a single amino acid, methionine. Since released methionine is detected, we can conclude that initiation is unaffected by oxydifficidin (Appendix A). Along with oxydifficidin, a positive control was chosen—Edeine A—an antibiotic that interferes with mRNA binding by overlapping with the codon location in the P-site and prevents the formation of the initiation complex, which is confirmed by the absence of ribosome stalling during toeprinting. Therefore, in this experiment, we do not visualize the release of BPy-Met.

#### 2.4.5. Competition for the Thiostrepton Binding Site

Recently, two spontaneous mutations (K84E and R76C) in the bL12 protein were shown to confer resistance to Oxy [21]. bL12 is a multi-copy component of the uL10/bL12 stalk of the 50S subunit: its flexible C-terminal domains recruit and stabilize the binding of ribosome-associated GTPases (e.g., EF-G, IF2, RF3), promote their activation and stimulate GTP hydrolysis, thereby facilitating initiation, elongation and termination of translation [46,47]. Thiostrepton (Appendix A) binds to ribosomal protein uL11 in the GTPase-associated center; its binding site is located in close proximity to the bL12 protein (Appendix A) [48]. We therefore investigated whether the binding sites of Oxy and Ths overlap on the ribosome.

To address this, we synthesized a fluorescent thiostrepton derivative, Ths-FITC (Appendix A), in which a fluorescein tag was attached to a truncated thiostrepton analog (truncThs) via the peptide linker βASGSGC to improve solubility. Conjugation of the FITC-βASGSGC peptide to the dehydroalanine residue of truncThs was performed by sulfa–Michael addition following a published procedure [49].

The resulting fluorescent derivative Ths-FITC bound to *E. coli* 70S ribosomes with high affinity (Appendix A), showing an apparent dissociation constant (K_D_) in the subnanomolar range. In a competitive binding assay, we evaluated Oxy’s ability to displace Ths-FITC from its ribosomal binding site (Figure 5). Unlike Ths and truncThs, which showed strong ribosome binding (K_D_ ≈ 0.26 and 0.21 nM, respectively—consistent with literature values for binding to a reconstituted complex of protein uL11 and 23S rRNA [49]), Oxy did not appreciably displace Ths-FITC.

Thus, oxydifficidin likely binds to a different site or even target, distinct from the Ths binding site that involves uL11 and helices H43/H44 of 23S rRNA.

#### 2.4.6. Oxydifficidin Does Not Affect Eukaryotic Translation or Cell Viability

Since oxydifficidin effectively inhibits prokaryotic translation, it was decided to test whether this antibiotic would inhibit translation in a cell-free system based on mammalian cell lysate HEK293. As for prokaryotic systems, different concentrations of Oxy were used. Oxydifficidin has been shown to have no effect on eukaryotic translation (Figure 4C).

Oxydifficidin and difficidin were not previously assessed for inhibiting in vitro translation in the eukaryotic system. In this study, we showed that Oxy does not affect eukaryotic translation in a wide range of concentrations, ending with a quite large 36 µM.

The cytotoxicity of Oxy was assessed for the immortalized human embryonic kidney cell line HEK293T. This line is widely known for its reliable and fast growth. For this line, the MTT assay was conducted in three independent replicas. The CC_50_—50% cytotoxicity concentration was chosen for evaluation of cytotoxicity. It is the concentration of the tested compound which reduce the cell viability by 50 percent. The test results revealed that the CC_50_ for Oxy was approximately 120 mg/L (123 ± 6 mg/L; ≈220 µM, Appendix A). This may possibly be linked to the structure of Oxy which is relatively rigid and contains polar groups on different sides of the molecule, which can hinder the ability to get through cell membranes.

## 3. Materials and Methods

### 3.1. Producents’ Characterization

#### 3.1.1. Collection, Isolation and Preservation

The oxydifficidin-producing strain EV17 was isolated from grapevine *Vitis* L. obtained via crossing European and American grape varieties *V. vinifera* × (*V. vinifera* + *V. labrusca* + *V. riparia* + *V. rupestris* + *V. berlandieri* + *V. aestivalis* + *V. cinerea*). The vineyard of ‘Moldova’, 26 years old, was located on the coast of the Azov Sea (Temryuk district, Krasnodarsky area). The variety belongs to the Euro-American genetic group and is an interspecific hybrid that is suitable for table use. This sort has a complex resistance to the main diseases of grapes—oidium (*Uncinula necator*), mildew (*Plasmopara viticola*), and gray rot (*Botrytis cinerea*).

The grapevine tissues (one-year shoots and mature stems) were collected in triple repetitions on 28 July 2021 and immediately transported to the laboratory. The samples (2.0 g) were surface-sterilized with 70% ethanol (1 min), then exposed in 5% NaClO (1 min) and trice rinsed with sterile distilled water. After drying at room temperature under aseptic conditions they were crushed in a sterile mortar with saline solution (10 mL of solution per 1 g of substrate) and 0.1 mL aliquots were transferred to MPA plates. After 3 days incubation at 28 °C, the grown endophytic bacteria were sampled and carefully transferred to a new sterile Petri dish for repeated cultivation. The long-term storage of the isolates was carried out as cell suspensions in LB media [50] with glycerol (20%, *v*/*v*) at −80 °C.

Another oxydifficidin-producing strain *B. velezensis* K-3618 was obtained from the collection of the All-Russian Research Institute of Agricultural Microbiology, ID RCAM 07246. Strain K-3618, an endophytic microorganism isolated from potato tubers of the Charoit variety, exhibits cellulolytic, amylolytic, and weak nitrogen-fixing activity.

#### 3.1.2. Cultivation

To obtain a sufficient amount of the active compound for detailed bioactivity studies, strain EV17 was cultured in four 750 mL Erlenmeyer flasks with 250 mL of liquid Organic medium 79 (g/L: glucose 10, peptone 10, yeast extract 2, hydrolysate casein 2, NaCl 6; pH 7.0, sterilized by autoclaving at 111 °C and 0.5 atm for 20 min) at 28 °C with shaking (200 rpm) for 3 days (OD_600_). Culture liquids were separated from biomass by centrifugation at 20,000× *g* for 20 min (Centrifuge 5810 R, Rotor FA-45-6-30, Eppendorf, Hamburg, Germany).

Strain K-3618 was initially grown on agar at 37 °C. A portion of the culture was transferred to a 750 mL Erlenmeyer flask containing 50 mL of 2YT medium (g/L: tryptone 16, yeast extract 10, NaCl 5; pH 7.0, sterilized by autoclaving at 121 °C and 1.0 atm for 20 min) and incubated at 28 °C for 24 h with shaking at 150 rpm on an Innova 40 thermostat shaker (New Brunswick Scientific, Edison, NJ, USA). The resulting culture was used as a 3% (*v*/*v*) inoculum to seed a second-generation culture in 150 mL of the same medium, which was cultivated under identical conditions within 72 h necessary for the culture to reach the stage of late stationary growth.

#### 3.1.3. Phenotypic Characterization

Cultural characteristics of EV17 and K-3618 were determined after incubation for 3 days at 28 °C on Organic media 79 and LB agar. Gram staining of cells was carried out using a Gram-reagent kit (OOO NICF, Saint Petersburg, Russia). Cell morphology was examined under light microscope (magnification ×1500) with oil immersion. Enzymes and carbon source utilization were investigated using paper discs to differentiate bacteria (Microgen, Moscow, Russia).

#### 3.1.4. Genome Sequencing and Annotation

The genomic data for the strain K-3618 were taken from the previous research [51].

The genome assembly was carried out with SPAdes v4.0.0 [52] in the “--careful” mode based on Illumina short reads corrected and quality-assessed with fastp v0.23.2 [53] and FastQC v0.12.1 [54], respectively, followed by quality check using QUAST v5.2.0 [55] and CheckM v1.2.2 [22]. Assembly coverage reached 99.37 reads per bp. Genome annotation was performed with the Prokka v1.14.6 [56] software with the custom database encompassing protein sequences from genomes of the *Bacillus* genus deposited in the NCBI RefSeq database [57].

To obtain a high-quality assembly of the strain EV17, both Il-lumina and Oxford Nanopore platforms, were employed. The LumiPure kit (Lumiprobe, Moscow, Russia) was used to isolate genomic DNA of EV17. A paired-end DNA library was prepared using the NEBNextUltra II DNA Library Prep Kit (Illumina, San Diego, CA, USA) and NEBNext Multiplex Oligos for Illumina (96 Unique Dual Index Primer Pairs) according to manufacturer instructions. Whole-genome sequencing was carried out by NovaSeq6000 sequencing platform (Illumina, San Diego, CA, USA) with 2  ×  250 bp read length using NovaSeq 6000 SP Reagent Kit v1.5 (500 cycles) according to the manufacturer’s protocol. Long genomic DNA reads were obtained by means of Oxford Nanopore Technologies (Oxford, UK). Fraction of high-molecular weight DNA was used for library preparation using the ONT Native Barcoding Kit V14 (SQK-NBD114) according to manufacturer instructions. Further sequencing was performed using R10.4.1 PromethION Flow Cell by means of ONT PromethION P2 sequencing platform, according to the manufacturer’s protocol (Oxford Nanopore Technologies, Oxford, UK). Nanopore sequencing of genomic DNA was performed using the ligation sequencing protocol SQK-LSK109 (Oxford Nanopore Technologies, Oxford, UK). Sample preparation followed the procedure described in reference [58]. Flow cell loading and library preparation were carried out according to the manufacturer’s instructions. The final DNA library was loaded onto an R9.4.1 MinION Mk flow cell (Oxford Nanopore Technologies, UK).

For multiplexed sequencing of several strains in a single run, a barcoding step was introduced prior to adapter ligation using the Native Barcoding Expansion 1–12 kit (EXP-NBD104, Oxford Nanopore Technologies, UK). Barcoding was performed according to the manufacturer’s protocol, which parallels the adapter ligation workflow and includes barcode ligation and DNA purification steps. Following barcoding ligation, equimolar DNA samples were pooled for the subsequent adapter ligation step.

Nanopore reads were filtered and trimmed with Nanofilt (v2.8.0) with defaults [59]. Raw Illumina sequencing reads were initially subjected to quality assessment using FastQC (v0.12.1) [54], and subsequent filtering of low-quality and adapter-contaminated reads was performed with fastp (v0.24.1) with defaults [53]. A hybrid assembly based on illumina and oxford nanopore reads was obtained using hybrid mode SPAdes with defaults [60]. To correct sequence errors in genomes assembled with long reads, Illumina reads and Nanopore assembly were used with NextPolish (v1.4.1) [61]. The construction of the phylogenetic tree was carried out using FastME v.2.1.1.6 analysis based on distance algorithms [34].

#### 3.1.5. Genome-Wide Taxonomy Classification

The genome sequence data were uploaded to the Type (Strain) Genome Server (TYGS), a free bioinformatics platform available under https://tygs.dsmz.de (accessed on 26 November 2025), for a whole genome-based taxonomic analysis [36].

Determination of closest type strain genomes was done in two complementary ways: First, all user genomes were compared against all type strain genomes available in the TYGS database via the MASH algorithm, a fast approximation of intergenomic relatedness [62], and, the ten type strains with the smallest MASH distances chosen per user genome. Second, an additional set of ten closely related type strains was determined via the 16S rDNA gene sequences. These were extracted from the user genomes using RNAmmer 1.2 [63] and each sequence was subsequently BLASTed https://blast.ncbi.nlm.nih.gov/Blast.cgi (accessed on 26 November 2025) [64] against the gene sequence encoded 16S rRNA of each of the currently 23,535 type strains available in the TYGS database. This was used as a proxy to find the best 50 matching type strains (according to the bitscore) for each investigated genomes and to subsequently calculate precise distances using the Genome BLAST Distance Phylogeny approach (GBDP) under the algorithm ‘coverage’ and distance formula d5 [65]. These distances were finally used to determine the 10 closest type strain genomes for each of the studied genomes.

For the phylogenomic inference, all pairwise comparisons among the set of genomes were conducted using GBDP and accurate intergenomic distances inferred under the algorithm ‘trimming’ and distance formula d5 [65]. 100 distance replicates were calculated each. Digital DDH values and confidence intervals were calculated using the recommended settings of the GGDC 4.0 [65,66]. The resulting intergenomic distances were used to infer a balanced minimum evolution tree with branch support via FASTME 2.1.6.1 including SPR postprocessing [34]. Branch support was inferred from 100 pseudobootstrap replicates each. The tree was rooted at the midpoint [35] and visualized with PhyD3 [67].

Information on nomenclature and synonymy was provided by TYGS’s sister database, the List of Prokaryotic names with Standing in No-menclature (LPSN, available at https://lpsn.dsmz.de (accessed on 26 November 2025)) [66]. The results were provided by the TYGS on 24 August 2025.

Average nucleotide identity analyses were calculated by using BLAST+ 2.11.0 using the JSpeciesWS server https://jspecies.ribohost.com/jspeciesws (accessed on 30 October 2025) [37]. ANI values were calculated to compare EV17 and K-3618 with their closest type strains with complete genome, established according to the previously described procedure.

The threshold criteria for determining whether a genome belongs to the same species as type strain genome are 70% and 95% similarity in DDH and ANI, respectively.

### 3.2. Purification and Isolation of Bioactive Compound

#### 3.2.1. Solid-Phase Extraction

Bacterial cells were removed from the culture broth by centrifugation on Sigma 3-16KL (Sigma Aldrich, Steinheim, Germany) at 5000 rpm followed by filtration through a 0.47 μm MCE membrane filter (Millipore, Burlington, MA, USA). One liter of the clarified supernatant was loaded onto a 30 mL cartridge containing 7 g of LPS-500-H polymer sorbent (divinylbenzene hydrophilic copolymer, pore size 50–1000 Å, 70 μm; Technosorbent, Moscow, Russia) at a flow rate of 15 mL/min using a peristaltic pump (Masterflex L/S Variable Speed Pump System, Masterflex, Gelsenkirchen, Germany). Sequential elution was performed with 15 mL portions of water-acetonitrile (MeCN) mixtures containing 0, 10, 35, 50, 75, and 100% MeCN. The biological activity of each fraction was assessed using the reporter *E. coli lptD^mut^* strain.

#### 3.2.2. HPLC Separation

The most active fraction, eluted at 75% MeCN, was further analyzed by reverse-phase HPLC using a Nexera X2 LC-30A system (Shimadzu, Kyoto, Japan) equipped with an SPD-M20A detector. Separation was performed on a Gemini NX C18 column (150 × 10 mm, 5 μm, 110 Å; Phenomenex, Torrance, CA, USA) with solvent A (10 mM NH_4_OAc, pH 5) and solvent B (MeCN). The elution profile consisted of isocratic elution at 40% solvent B for 15 min, followed by a gradient to 90% MeCN over 17 min and a 3 min column wash; flow rate was 3 mL/min and detection was at 275 nm. Fractions were collected and assayed for activity; the fraction containing the pure active compound was isolated and subjected to LC-MS analysis.

For biological assays, the active compound was purified on a semipreparative Gemini NX C18 column (150 × 20 mm, 10 μm, 110 Å; Phenomenex) using a PuriFlash 5.250 system (Interchim, Montluçon, France) under the same solvent conditions, with a flow rate of 16 mL/min.

### 3.3. Identification of Bioactive Compound

#### 3.3.1. Mass-Spectrometry

LC-MS analysis was carried out on an Ultimate 3000 RSLCnano HPLC system connected to an Orbitrap Fusion Lumos mass spectrometer (ThermoFisher Scientific, Waltham, MA, USA) with the loading pump used for analytical flow gradient delivery. Samples were separated on a Gemini NX-C18 3 μm 100 Å column 100 × 2.1 mm at 200 μL/min flow rate in the linear gradient of acetonitrile in water with the addition of 10 mM ammonium formate and 0.1% formic acid. UV data were collected at 220 and 280 nm. MS1 and MS2 spectra were recorded at 30 K and 15 K resolution, respectively, with HCD fragmentation. Raw data were collected and processed on Thermo Xcalibur Qual ver. 4.3.73.11.

#### 3.3.2. NMR Spectroscopy

NMR spectra were acquired on a Bruker AVANCE spectrometer operating at 600.13 MHz for 1H and 242.93 MHz for 31P. Measurements were performed at 298 K using CD3 OD as the solvent. 1H chemical shifts are reported relative to TMS at 0 ppm. 31P chemical shifts are referenced to triethyl phosphite (139.9 ppm), which was used as an internal standard [68]. All spectra were processed and analyzed using Mestrelab Mnova 16.0.0 software (Mestrelab Research, Santiago de Compostela, Spain).

#### 3.3.3. Analysis of Antibiotic’s Biosynthetic Gene Clusters

Secondary metabolite biosynthetic gene clusters in the complete genome of strains EV17 and K-3618 were identified with the bacterial version of antiSMASH 8.0 [69] (https://antismash.secondarymetabolites.org/ (accessed on 30 October 2025)).

### 3.4. Biological Activity Testing

#### 3.4.1. Design of Reporter Strain *E. coli Lptd^mut^* pDualrep2.1

The previously developed *E. coli* Δ*tolC* pDualRep2 system [38] lacks active efflux of toxic compounds, including antibiotics. However, its outer membrane remains intact, limiting the penetration of larger or more hydrophobic molecules.

To overcome this limitation and further enhance compounds permeability, the *E. coli lptD^mut^* strain was developed. In this strain, a deletion of 69 nucleotides in the *lptD* gene disrupts lipopolysaccharide assembly in the outer membrane, increasing membrane permeability. Similar to the *E. coli* Δ*tolC* pDualRep2 system, *E. coli lptD^mut^* expresses fluorescent proteins under the same operon control and regulatory mechanisms.

Initially, it was anticipated that the *E. coli* BW25113 *lptD^mut^* strain could be transformed with the standard AmpR pDualRep2 plasmid [38]. However, the transformed cells exhibited instability under ampicillin selection and survived only at concentrations up to 25 μg/mL. To overcome this limitation, the ampicillin resistance cassette of pDualRep2 was replaced with a kanamycin resistance cassette.

The vector backbone was obtained from pDualRep2, and the kanamycin resistance gene was amplified from the tolC mutant of the KEIO collection [70] using primers pdualrep2_fwd/pdualrep2_rev and KanR_fwd/KanR_rev, respectively (Appendix A). PCR products were purified with the Cleanup Mini Kit (Qiagen, Hilden, Germany) and verified on 1% agarose gel electrophoresis. DNA fragments were assembled using NEBuilder^®^ HiFi DNA Assembly (New England Biolabs, Ipswich, MA, USA) according to the manufacturer’s protocol. The assembled products were size-selected on agarose gel and purified with the Cleanup Mini Kit.

For transformation, 1 mL of overnight *E. coli* BW25113 *lptD^mut^* culture (OD600 = 2.0) was chilled on ice for 10 min, pelleted by centrifugation at 5000 rpm for 10 min, washed twice with TB buffer (10 mM PIPES, 15 mM CaCl_2_, 250 mM KCl, 55 mM MnCl_2_, pH 6.7), and resuspended in 1 mL of TB. Aliquots (100 μL) of competent cells were mixed with 100 ng of pDualRep2-KanR plasmid DNA and incubated on ice for 30 min. Heat shock was performed at 42 °C for 45 s, followed by recovery in 800 μL LB medium at 37 °C for 1 h. Transformants were plated on LB agar containing kanamycin (50 µg/mL) and incubated overnight.

A single colony of *E. coli* BW25113 *lptD^mut^* harboring pDualRep2-KanR was inoculated into 50 mL LB medium supplemented with kanamycin and cultured at 37 °C with shaking (100 rpm) for 15 h. The culture was supplemented with sterile glycerol to a final concentration of 25% (*v*/*v*), aliquoted (0.5 mL), snap-frozen in liquid nitrogen, and stored at −20 °C.

#### 3.4.2. Reporter Antibacterial Assays on Agar Plates

The two *E. coli* reporter strains: *E. coli lptD^mut^* pDualrep2.1 and JW5503 Δ*tolC* pDualRep2 were used in this work as previously described [71]. Briefly, the overnight culture of reporter strains was diluted with fresh LB medium to an optical density of 600 nm (OD600) of 0.05–0.1. The culture was transferred to LB agar plates that had 100 μg/mL ampicillin or 50 μg/mL kanamycin applied for JW5503 Δ*tolC* pDualRep2 and *E. coli lptD^mut^* pDualrep2.1 strains, respectively. On an agar plate with the lawn of one of the reporter strains 10 µg of oxydifficidin was applied along with two control antibiotics: erythromycin (Ery, 5 mg/mL) and levofloxacin (Lev, 25 mg/mL). Plates were incubated at 37 °C overnight and then scanned by ChemiDoc (Bio-Rad, Tokyo, Japan) in the modes ‘Cy3-blot’ for RFP and ‘Cy5-blot’ for Katushka2S. In the case of SOS-response activation the expression of the *rfp* gene occurred, while the expression of *katushka2S* gene took place in the case of a violation of translation, when the ribosome was stalled on the mRNA template. When scanning, the signal from the RFP protein was displayed in green, and from Katushka2S in red.

#### 3.4.3. MIC Determination

Overnight cultures of tested strains were diluted 1:1000 in LB medium. A two-fold serial dilution was then carried out. The ninety-six-well 2 mL deep-well plates containing *E. coli* KanR culture with and without Oxy, and LB medium as a control, along with Ery, which was used as a control for the experiment, were then incubated overnight at 37 °C with shaking at 200 rpm. Cell growth was measured at 590 nm using a microplate reader (VICTOR X5 Light Plate Reader, PerkinElmer, Waltham, MA, USA). MIC determination was performed in three independent biological replicates, each consisting of two parallel technical replicates (see Data Availability Statement).

#### 3.4.4. Bacterial In Vitro Translation Assay

To test the ability of Oxy to inhibit protein synthesis in vitro the PURExpress^®^ In Vitro system (NEB, Ipswich, MA, USA) or *E. coli* S30 Extract System for Linear Templates (Promega, Madison, WI, USA) was used. The assembled reactions (5 μL) were supplemented 0.1 mM of d-luciferin (Promega), 0.5 μL of either antibiotic solution or water, and 100 ng of *Fluc* mRNA. All samples were then placed in a 384-well black-wall plate at 37 °C. Chemiluminescence was recorded with VICTOR X5 Light Plate Reader. The Fluc mRNA obtained by MEGAscript™ T7 Transcription Kit (ThermoFisher, Carlsbad, CA, USA) from the circular DNA template.

#### 3.4.5. Toeprinting Assay

Toeprinting was carried out according to the protocol described in [43] with minor modifications. Toeprinting reactions were carried out in 5 μL aliquots containing 2 μL of solution A, 1 μL of solution B (PURExpress transcription-translation coupled system (New England Biolabs, USA)), 0.2 μL of RiboLock (ThermoFisher), 0.5 μL of the oxydifficidin (final concentrations 36, 3.6 and 0.9 µM), 0.5 μL of DNA template *ermDL* or *ermCL* (100 ng), and 0.5 μL of the 5′-end [^32^P]-radiolabeled (1 pmol) or [FAM]-labeled (10 pmol) NV1 primer. The reactions were incubated at 37 °C for 20 min. Reverse transcription was conducted for 15 min at 37 °C using AMV Reverse Transcriptase (New England Biolabs, USA). The reaction was then stopped by adding 1 μL of 10 M NaOH (15 min at 37 °C), neutralized by 1 μL 10 N HCl and purified by QIAquick PCR purification kit (Qiagen, Germany). Primer extension products were resolved on 6% polyacrylamide gel containing aqueous 19:1 solution of acrylamide and N,N′-methylenebisacrylamide and 7 M urea in TBE buffer. Results were visualized using a Typhoon FLA 9500 Biomolecular Imager (GE Healthcare, Chicago, IL, USA). The experiments were conducted in triplicate.

The *ermDL*, *ermCL* template and NV1 primer sequences described in Appendix A.

#### 3.4.6. Fluorescently Labeled Short Peptides

Coupled transcription-translation was set up in 5 µL reactions using a PURExpress Δ (aa, tRNA) Kit (NEB, Ipswich, MA, USA) as described previously [45] with minor modifications: 20 ng of MF2 DNA template containing a T7 promoter upstream of the coding sequence, 0.1 µM BPY-Met-tRNA^fMet^, 0.45 µM fMet-tRNA^fMet^, 1 µM Phe-tRNA^Phe^, 100 µM Phe and 1 μL of either antibiotic solution or water were added to each reaction.

To assess the effect of oxydifficidin on translation initiation the experiment was carried out in a system of PURExpress^®^ In Vitro Protein Synthesis Kit using template that encodes one amino acid—methionine (M template). The reactions were divided into two parts: with and without treatment by RNAse A (ThermoFisher) (15 min incubated on ice). The rest of the protocol was not subject to further changes.

Samples then were preheated for 3 min at 70 °C and loaded to a 10% denaturing PAGE (19:1 AA:bisAA; 1× TBE buffer; 7M urea). Gels were scanned by a Typhoon FLA 9500 Biomolecular Imager (GE Healthcare) in the FAM channel with excitation peak (493 nm) and emission peak (517 nm).

#### 3.4.7. Mammalian Cell-Free System

Whole home-made HEK293T cell extracts were used to test compounds in a mammalian in vitro translation system. The reaction was carried out in 10 µL, including 5 µL HEK293T extract, 1 µL 10X translation buffer (20 mM Hepes-KOH pH 7.6, 1 mM DTT, 0.5 mM spermidine-HCl, 0.8 mM Mg(OAc)_2_, 8 mM creatine phosphate, 1 mM ATP, 0.2 mM GTP, 120 mM KOAc and 25 μM of each amino acid), 2U of RiboLock RNase inhibitor (Thermo Scientific), 0.5 mM d-luciferin (Promega), 1 μL of either antibiotic solution or solvent (water), and 50 ng mRNA (the latter was added to 1 μL of the mixture solution after preliminary incubation of the reaction mixture with the antibiotic for 5 min at 30 °C.). After adding the mRNA, the mixtures were transferred to a pre-heated white FB/NB 384 well plate (Grenier no. 781904) and incubated in the VICTOR X5 Multilabel Plate Reader (PerkinElmer, Waltham, MA, USA) at 30 °C with continuous measurement of luciferase activity.

#### 3.4.8. Competition for the Thiostrepton Binding Site

The fluorescence anisotropy method was employed to assess competition at the thiostrepton binding site on the ribosome. 70S ribosomes were purified from *E. coli* MRE600 cells following a published procedure [72]. The fluorescent thiostrepton derivative Ths-FITC was synthesized as previously described [49], with detailed procedures provided in the Appendix A. Binding of Ths-FITC to *E. coli* 70S ribosomes was assessed by incubating 4 nM Ths-FITC with ribosomes (0.1–200 nM) for 2 h at 25 °C in buffer containing 20 mM HEPES-KOH (pH 7.5), 50 mM NH_4_Cl, 10 mM Mg(CH_3_COO)_2_, and 0.05% Tween-20. Binding affinities of Ths, its truncated derivative truncThs, and Oxy for the *E. coli* ribosome were determined by a competition-binding assay with Ths-FITC (4 nM) and ribosomes (7.3 nM) in the buffer. Test compounds, initially dissolved in 2,2,2-trifluoroethanol (Ths and truncThs) or DMSO (Oxy), were added to pre-formed complexes at final concentrations ranging from 0.5 nM to 100 μM, ensuring that the concentration of organic solvent did not exceed 5%. The mixtures were then incubated for 4 h at 25 °C. A 5% DMSO solution, corresponding to the highest solvent concentration used in the assay, served as a negative control. Fluorescence anisotropy was measured on a VICTOR X5 Multilabel Plate Reader (PerkinElmer, Waltham, MA, USA) using a 384-well format (excitation wavelength was 485 nm, and the emission wavelength was 535 nm). All measurements were performed in quadruplicate. Apparent dissociation constants were calculated as described [73].

#### 3.4.9. MTT Cytotoxicity Test

The impact of the test compound on cellular metabolic activity, as an indicator of cell viability, was evaluated using MTT reduction assay as described [74]. Investigations were performed solely on the human embryonic kidney HEK293T cell line. Solvent DMSO served as the reference control compound. Oxy was prepared as DMSO stock solution with a concentration 40 mg/mL.

HEK293T cells were cultured in DMEM/F-12 medium, enriched with 10% FBS and a 1% antibiotic-antimycotic solution (penicillin 50 U/mL, streptomycin 50 µg/mL), under standard conditions (37 °C, 5% CO_2_). For experimental procedures, cells were plated in triplicate at 2500 cells/well in 96-well plates and incubated for 24 h to ensure attachment. Post-attachment, the culture medium was supplemented with serial two-fold dilutions of the compounds; the concentration range for the Oxy was 2–250 μg/mL.

The treated cells were incubated for 72 h. Thereafter, the MTT reagent was introduced to a final concentration of 0.5 g/L and the plates were incubated for an additional 2 h to allow formazan crystal formation. The supernatant was subsequently removed, and the crystals were fully dissolved by adding 140 µL of DMSO and agitating the plates for 10 min. The optical density of the resultant solution was quantified at a wavelength of 555 nm utilizing a microplate reader (VICTOR X5 Multilabel Plate Reader (PerkinElmer, Waltham, MA, USA)). Data normalization and subsequent analysis, including the derivation of IC_50_ values from dose–response curves and the calculation of standard errors, were performed employing GraphPad Prism version 8.0.2. (GraphPad Software, Inc., San Diego, CA, USA).

## 4. Conclusions

In conclusion, this study confirms the significant antibiotic potential of oxydifficidin, a natural polyketide known to target the bacterial ribosome. We identified two producer strains, *B. velezensis* EV17 and K-3618 from distinct climate zones, and employed a toeprinting assay to demonstrate that oxydifficidin induces a generalized arrest of protein synthesis by impeding multiple stages of the translation process. At the same time, it does not affect the initiation of translation.

While prior research established the bL12 protein as a target, our findings provide a critical refinement: the binding site of oxydifficidin is distinct and does not overlap with that of the canonical translation inhibitor, thiostrepton. Furthermore, a key advantage of oxydifficidin is its selective action, as it exhibits no inhibitory effect on eukaryotic translation machinery and shows a two orders of magnitude lower effect on eukaryotic cells compared to bacteria.

Collectively, these findings—its novel mechanism of translational inhibition, coupled with its selectivity for prokaryotic systems—underscore the promise of oxydifficidin for future development. It presents a compelling candidate for applications as an effective bioprotectant against phytopathogens in agriculture and warrants further investigation for its potential therapeutic use.

## Figures and Tables

**Figure 1 ijms-26-11777-f001:**
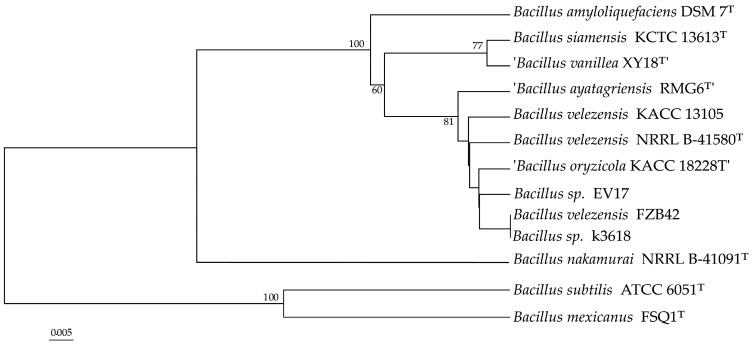
Tree inferred with FastME 2.1.6.1 [34] from GBDP distances calculated from genome sequences. The branch lengths are scaled in terms of GBDP distance formula d5. The numbers above branches are GBDP pseudo-bootstrap support values > 60% from 100 replications, with an average branch support of 56.5%. The tree was rooted at the midpoint [35]. The names of species with unconfirmed nomenclatural status are given in quotation marks.

**Figure 2 ijms-26-11777-f002:**
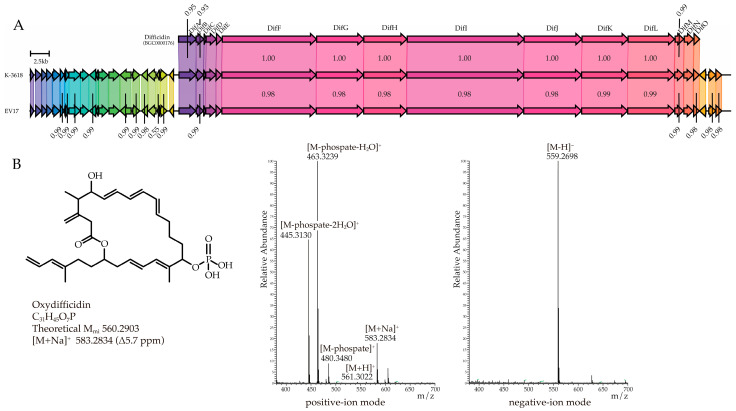
(**A**) Analysis of bioactive metabolites from the strains EV17 and K-3618: a comparison of difficidin BGC (BGC0000176) with regions 3 in the EV17 and K-3618 complete genome sequence, generated using clinker tool [42]. All three BGCs harbor a highly similar biosynthetic gene cluster (EV17’s and K-3618’s clusters have 98.9% and 100% identity with BGC0000176, respectively), correlation values not indicated in the figure are 1.00; additionally, EV17 and K-3618 genomes exhibit extended conserved synteny of flanking regions (both upstream and downstream of the cluster), indicating a shared genomic context beyond the core locus. Homologous genes are highlighted with colors, and labels indicate identity of the genes. (**B**) Structure and HR-LCMS spectra of the isolated oxydifficidin.

**Figure 3 ijms-26-11777-f003:**
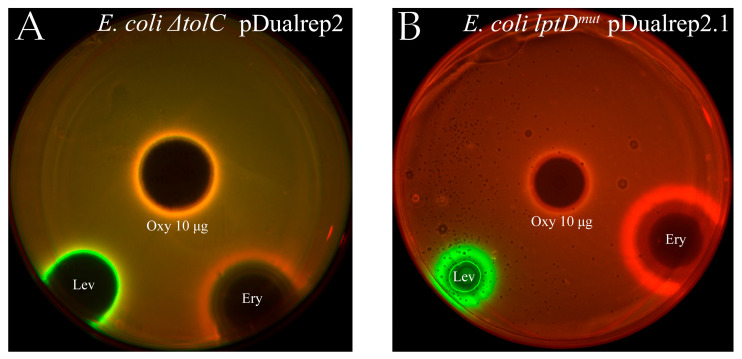
Oxydifficidin inhibits protein synthesis in bacteria cells. Agar plates coated with (**A**) *E. coli* Δ*tolC* pDualrep2 [38] and (**B**) *E. coli lptD^mut^* pDualrep2.1 reporter strains and spotted with oxydifficidin (10 µg) along with erythromycin (Ery) (5 mg/mL) and levofloxacin (Lev) (25 mg/mL). The plates were scanned in Cy3 (for TurboRFP) and Cy5 (for Katushka2S) channels, shown as green and red pseudocolor, respectively.

**Figure 4 ijms-26-11777-f004:**
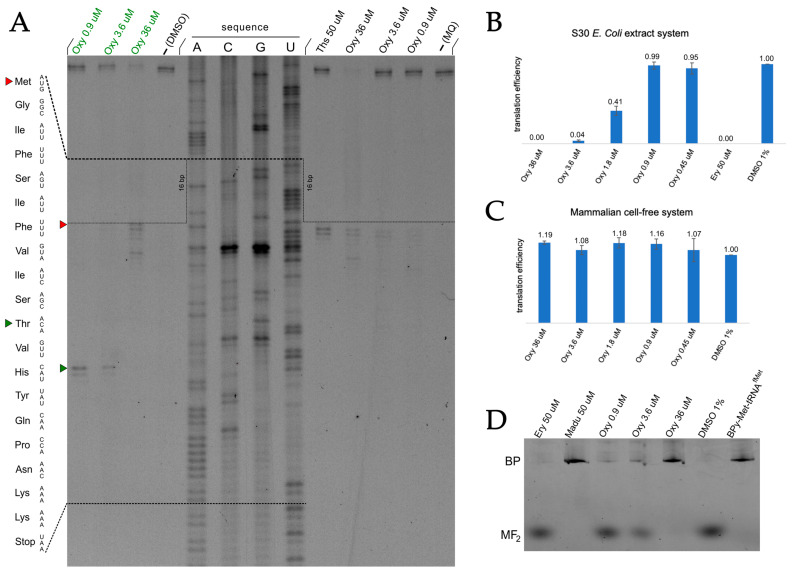
Oxydifficidin inhibits prokaryotic translation, but not eukaryotic. (**A**) Oxy causes non-specific pauses in translation distributed across multiple codons. Toeprinting assay of *ErmCL* template in the presence or absence (−) of Oxy at various concentrations with (highlighted in green) or without Borr (highlighted in black) and control antibiotic thiostrepton (Ths). Ths inhibits initiation (red arrow), Borr stalls translation on threonine codon (green arrow), while Oxy did not cause ribosome stalling at specific mRNA sites but instead caused non-specific pauses distributed across nearly every codon in the first part of the transcript. (**B**) In vitro translation in the *E. coli* S30 extract system. Oxydifficidin inhibits prokaryotic translation in a concentration-dependent manner. Positive control—erythromycin (50 µM), negative control—1% DMSO. The graph represents means of three independent replicates; error bars indicate ± SD. (**C**) In vitro translation in the mammalian cell-free HEK293 lysate system. The graph represents means of three independent replicates; error bars indicate ± SD. (**D**) The products of MF2-coding mRNA in vitro translation in the presence of BPY-Met-tRNA^fMet^ and antibiotics indicated above the lanes: oxydifficidin 0.9 µM, 3.6 µM and 36 µM concentrations, erythromycin 50 µM, madumicin 50 µM, along with control BODIPY-Met-tRNA^fMet^ itself (BPy) and sample without antibiotics as a negative control (DMSO 1%). BP stands for BODIPY label; MF_2_ stands for tripeptide (Met-Phe-Phe) product.

**Figure 5 ijms-26-11777-f005:**
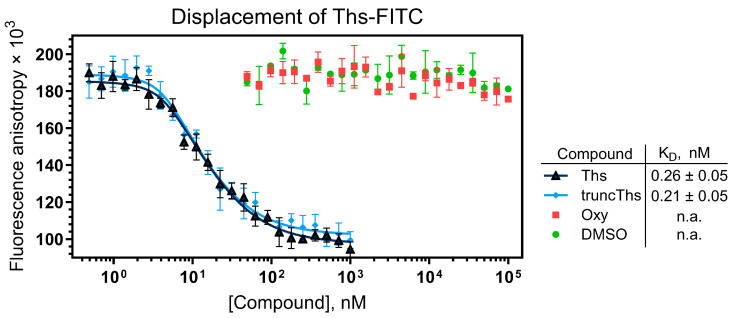
Oxydifficidin does not compete for the thiostrepton binding site. A competitive binding assay measuring the ability of Ths, its truncated analog (truncThs), and Oxy to compete for the thiostrepton binding site on *E. coli* 70S ribosomes, using fluorescence anisotropy of fluorescently labeled truncated thiostrepton (Ths-FITC). A 5% DMSO solution was used as a control. Data represent means of four independent replicates; error bars indicate ±SD. The apparent dissociation constants (K_D_) with CI (α = 0.05) are shown. n.a.—not applicable.

**Table 1 ijms-26-11777-t001:** Comparative analysis of genomes of *B. velezensis* strains.

	EV17	FZB42 [25]	NRRL B-41580^T^ [26]	K-3618
General features				
DNA GC content	46.5%	46.5%	46.3%	46.4%
Genome size (bp)	3,978,967	3,918,589	4,034,335	3,864,632
Protein CDS	3829	3693	3790	3734
Extracellular carbohydrate degrading enzymes				
Amylase AmyE	98.23%	100%	96.41%	100%
Non-ribosomal synthesis of lipopeptides and polyketide				
Bacillaene (GenBank ID: AJ634060.2)	98.68%	100%	98.07%	100%
Fengycin (AJ576102.1 ^1^)	98.36%	100%	97.92%	Low coverage
Macrolactin H (AJ634061.2)	98.83%	100%	98.2%	100%
Difficidin (AJ634062.2)	98.64%	100%	98.06%	100%

^1^ *B. velezensis* partial genome, strain FZB42, containing *fenE* gene responsible for fengycin synthesis.

**Table 2 ijms-26-11777-t002:** The characteristics and comparison of the whole genome of EV17 and K-3618 with their closest *B. velezensis* strains.

Strain	dDDH for Strains		ANI [37] for Strains		Diff. DNA GC Content		Accession No.
	EV17	K-3618	EV17	K-3618	EV17	K-3618	
EV17	--	90.3%	--	98.82%	--	0.13%	CP199744.1
K-3618	90.3%	--	98.82%	--	0.13%	--	GCA_050472105.1
*B. velezensis* FZB42	90.4%	100%	98.77%	99.99%	0.05%	0.08%	GCA_000015785
*B. velezensis* NRRL B-41580^T^	85.8%	85.8%	98.19%	98.25%	0.21%	0.08%	GCA_001461825
*B. velezensis* KACC 13105	84.9%	85.1%	98.13%	98.16%	0.1%	0.03%	GCA_000960265

## Data Availability

The original contributions presented in this study are included in the article/Appendix A. Further inquiries can be directed to the corresponding author.

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
