# Peer review of "Characterization of Bacillus velezensis EV17 and K-3618 and Their Polyketide Antibiotic Oxydifficidin, an Inhibitor of Prokaryotic Translation with Low Cytotoxicity"

_ijms, 2025, doi:10.3390/ijms262411777_

Round 1

Reviewer 1 Report

Comments and Suggestions for Authors

In the manuscript titled “Insights into the mechanism of action of antibiotic oxydifficidin”, Chernyshova and colleagues explore the mechanism of action of an “abandoned” antibiotic oxydifficidin, discovered more than 35 years ago, using an array of biochemical methods. However, unfortunately, the results presented do not significantly expand our understanding of the drug’s MOA compared to what we already knew from previously published works. Several results are negative (lack of overlap with the binding site of thiostrepton) or unrelated to the mode of translation inhibition (biosynthesis by certain strains of Bacillus), while the interpretation of the key biochemical assays raises multiple questions.

  • Toe-printing assay (Figure 4A):
    1. In the presence of DMSO, multiple stalls are observed in the template, with overall intensity similar to that observed upon the addition of Oxy, suggesting potential intrinsic propensity of the selected template for translation slow-down/stalling under the tested conditions. How can the authors distinguish between Oxy- and DMSO-induced stalling? Why did they not use any “catch codon” to reveal the amount of residual translation reaching a particular point/end of the template?
    2. The authors use two concentrations of the drug differing by 10 times, for both of which they show 100% inhibition of protein synthesis in S30 lysate. The pattern of bands and their intensity is very similar in the toe-printing assay; however, in panel 4D, the lower of the two concentrations of Oxy allows for the synthesis of a considerable amount of the 3 amino acid-long peptide. How can this be explained, given the use of the kits with comparable ribosome concentrations for both assays?
    3. The authors mention a “strong band corresponding to inhibition at the initiation step” for the sample with Ths added; however, the band’s intensity insignificantly differs from that of the initiation band for the DMSO-containing sample.
    4. The overall assay quality is poor (probably the [32P]-ATP used was not fresh enough), and the bands are very faint. Apparently, the assay was performed only once (I could not find the number of replicates in the corresponding section of the Methods or any supplementary figures related to this assay), and only one template was tested.

In my opinion, the data presented for this assay lacks the quality required to substantiate the authors' conclusions.

  • Assays with fluorescently labelled peptides:
    1. Figure 4D lacks a proper positive control; there is no antibiotic with an established mechanism of action with which we could compare the result for Oxy. Erythromycin expectedly did not inhibit the reaction due to the lack of a +X+ motif in the nascent peptide MFF.
    2. Interpretation of the results in Figure S5 is confusing to me as well: the authors state that “Since released methionine is detected, we can conclude that initiation is unaffected by oxydifficidin”, while the patterns observed on the gel for Ths (an established initiation inhibitor) and Oxy do not differ.

  • FP-based assays: as far as I could get from the Methods, pDualrep2 and pDualrep2.1 differ only by the resistance marker used (AmpR vs KanR). Why do the assays using these two plasmids (Figure 3) produce different signals (Red and Green pseudocolors) for the same control antibiotic Ery? How do the authors explain the same signal detected for Lev and Ery for pDualRep2.1?

In addition, the work is difficult to read; the Results section contains very technical parts more appropriate for the Methods section (e.g., lines 182-186 and the whole part 2.2). The work lacks a coherent narrative (e.g., the assay with fluorestent protein Katushka2S is first used in section 2.2, while it is explained only later in 2.4.1).

To sum up, the work does not present a clearly substantiated advancement regarding the mode of action of oxydifficidin beyond previously established findings. As such, in my opinion, the manuscript does not meet the novelty and mechanistic insight standards for publication in the International Journal of Molecular Sciences.

Author Response

We thank the reviewer for their valuable feedback on our manuscript. In response to their comments, we have performed additional experiments, including a toe-printing assay and a short-peptide synthesis assay, and have further evaluated oxydifficidin using our reporter system. The manuscript has been substantially revised to incorporate these new findings and address the points raised. Our point-by-point responses are provided below.

Comment 1: In the manuscript titled “Insights into the mechanism of action of antibiotic oxydifficidin”, Chernyshova and colleagues explore the mechanism of action of an “abandoned” antibiotic oxydifficidin, discovered more than 35 years ago, using an array of biochemical methods. However, unfortunately, the results presented do not significantly expand our understanding of the drug’s MOA compared to what we already knew from previously published works. Several results are negative (lack of overlap with the binding site of thiostrepton) or unrelated to the mode of translation inhibition (biosynthesis by certain strains of Bacillus), while the interpretation of the key biochemical assays raises multiple questions.

Response 1: Thank you for your careful review of our work and for the insightful comments. Our study indeed focuses not only on elucidating the mechanism of action of oxydifficidin but also on investigating the Bacillusstrains that produce it—important biocontrol agents effective against phytopathogens. Re-evaluating previously overlooked antibiotic is a central goal of this study. Considering that most antibacterial agents are natural products, systematizing knowledge about the bacterial strains that produce them is of particular importance.

Beyond characterizing the mode of action, our work also addresses key requirements that define a promising antibiotic. We were the first to demonstrate that oxydifficidin exhibit cytotoxicity levels two orders of magnitude lower than MIC against bacterial cells and that it does not inhibit eukariotic translation. These findings underscore the potential of the oxydifficidin chemical scaffold as a promising antibiotic candidate.

Considering mode of action itself, we were also the first to show that oxydifficidin does not inhibit initiation complex formation, as demonstrated using short fluorescent labelled peptides, but instead acts by inhibiting elongation.

To reflect this broader scope of our study, we have accordingly revised the title of the article to:«Characterization of Bacillus velezensis EV17 and K-3618 and Their Polyketide Antibiotic Oxydifficidin, an Inhibitor of Prokaryotic Translation with Low Cytotoxicity»

  • Comment 2: Toe-printing assay (Figure 4A):
    1. In the presence of DMSO, multiple stalls are observed in the template, with overall intensity similar to that observed upon the addition of Oxy, suggesting potential intrinsic propensity of the selected template for translation slow-down/stalling under the tested conditions. How can the authors distinguish between Oxy- and DMSO-induced stalling? Why did they not use any “catch codon” to reveal the amount of residual translation reaching a particular point/end of the template?

Response 2: Thank you for your comment. Considering that DMSO-induced stalling is more pronounced at the first codon and at the leucine codon—further downstream than the last visible stops observed with oxydifficidin—we can conclude that DMSO-related stalling reflects the inherent complexity of translation initiation and the influence of mRNA secondary structures. In contrast, oxydifficidin exhibits a more generalized stalling pattern, producing stops at every codon in the initial portion of the transcript.

To better distinguish between Oxy- and DMSO-induced ribosome stalling, we redesigned the experiment using the ErmCL template, which contains a threonine codon that can serve as a “catch codon.” Borrelidin was used as a control antibiotic for threonine-specific stalling. Due to limited access to fresh radioactive material, the experiment was reproduced using fluorescently labeled substrates. Nevertheless, the results confirmed a concentration-dependent inhibition of translation: at the highest oxydifficidin concentration (36 µM), the band corresponding to the “borrelidin-stalling” was barely detectable, whereas at the lowest concentration (0.9 µM), residual translation reaching the threonine codon was observed, reflected by a broader band than that of 3.6 µM Oxy, indicating a concentration-dependent manner of inhibition.

    1. Comment 3: The authors use two concentrations of the drug differing by 10 times, for both of which they show 100% inhibition of protein synthesis in S30 lysate. The pattern of bands and their intensity is very similar in the toe-printing assay; however, in panel 4D, the lower of the two concentrations of Oxy allows for the synthesis of a considerable amount of the 3 amino acid-long peptide. How can this be explained, given the use of the kits with comparable ribosome concentrations for both assays?

Response 3: Thank you for this valuable comment. We acknowledge that the initial version of the manuscript did not include a sufficiently detailed discussion of the results. Moreover, the conclusions drawn are not based on a simple “one experiment–one conclusion” approach, but rather arise from the integration of multiple complementary assays that together provide a more comprehensive interpretation of the data. Sections 2.4.2-2.4.4. has now been substantially revised to provide clearer interpretation and context for the findings.

As shown by the toeprinting assay, oxydifficidin causes general inhibition of translation, with multiple bands appearing throughout the transcript, indicating that portions of the template are translated. This cumulative inhibitory effect becomes less apparent on shorter templates, which can still support the synthesis of a complete, albeit short, translation product. Moreover, in the toeprinting assay, the visualized signal corresponds to DNA, the product of reverse transcription, whereas in the fluorescently labeled short peptide assay, the peptides themselves are directly detected. This difference in the nature of the readout makes a direct, one-to-one comparison of visualized bands between the two assays somewhat inappropriate.

These observations are consistent with the complete inhibition of protein synthesis observed in the 30S lysate assay at comparable concentrations, as that assay requires full-length product formation to generate a chemiluminescent signal. Based on oxydifficidin’s pattern of repeated binding and dissociation from the ribosome during elongation, observed in our initial toeprinting experiments, we used higher concentrations in an attempt to identify specific stalling sites; however, no motif-specific stalling was detected.

    1. Comment 4: The authors mention a “strong band corresponding to inhibition at the initiation step” for the sample with Ths added; however, the band’s intensity insignificantly differs from that of the initiation band for the DMSO-containing sample.

Response 4: Thank you for this comment. We agree that the phrase “strong band” was an unfortunate choice for describing this particular result. Conceptually, however, the interpretation remains correct. Translation initiation is one of the most complex and time-consuming stages of protein synthesis. Even in the absence of antibiotics, some ribosomes naturally stall at this stage—as well as at several codons along the transcript—due to mRNA secondary structures and other inherent biological factors. These events are detectable during reverse transcription, as also observed in the DMSO control. Thiostrepton, an inhibitor of initiation and translocation, produces a distinct stop exclusively at the first codon. To avoid any potential misunderstanding, the phrase “strong band” has been changed to “distinct stop” in the main text of the manuscript for greater clarity.

    1. Comment 5: The overall assay quality is poor (probably the [32P]-ATP used was not fresh enough), and the bands are very faint. Apparently, the assay was performed only once (I could not find the number of replicates in the corresponding section of the Methods or any supplementary figures related to this assay), and only one template was tested.

Response 5: We appreciate the reviewer’s concern regarding the reproducibility and the quality of the assay. The experiment was in fact repeated multiple times (three biological replicates using five independent templates), as now clarified in the Methods section. Since oxydifficidin did not produce any motif-specific stops on any of the tested templates, we chose to include in the manuscript the most representative template, ErmDL, as it contains a diverse range of amino acid codons.

Regarding the use of [γ-32P]-ATP, we acknowledge that the isotope available at the time was not fresh. Due to limited access to new radioactive material, we reproduced the experiment using fluorescently labeled substrates. While fluorescence-based toeprinting is generally less sensitive than the radioactive method, the overall pattern and quality of the results were consistent with those obtained previously, confirming the reproducibility and reliability of the assay.

Comment 6: In my opinion, the data presented for this assay lacks the quality required to substantiate the authors' conclusions.

Assays with fluorescently labelled peptides:

    1. Figure 4D lacks a proper positive control; there is no antibiotic with an established mechanism of action with which we could compare the result for Oxy. Erythromycin expectedly did not inhibit the reaction due to the lack of a +X+ motif in the nascent peptide MFF.

Response 6: Dear reviewer, we redesigned the experiment with BODIPY-labeled fluorescent peptides and included the positive control - madumycin (50 uM) that is the well-known elongation inhibitor. Madumycin blocks the formation of the first peptide bond. Therefore, on the gel we observe the original band from BPY-Met-tRNAfMet. Not all antibiotics are capable of blocking the biosynthesis of BODIPY-MF2, so erythromycin was chosen as a negative control antibiotic, which does not inhibit protein translation due to the absence of the +X+ motif. Oxydifficidin at a high concentration (36 uM) completely inhibits the formation of the full-length product - BODIPY-MF2 (only the original band corresponding to BODIPY-Met-tRNAfMet is visualized), at concentrations of 3.6 and 0.9 uM of oxydifficidin suppression of synthesis does not occur due to the insufficient number of antibiotic molecules. Noticeably, intermediates such as BODIPY-M or BODIPY-MF (which can be formed upon antibiotic exposure) are not visualized in this gel because they remain bound to the tRNA in the A and P sites of the ribosome.

    1. Comment 7: Interpretation of the results in Figure S5 is confusing to me as well: the authors state that “Since released methionine is detected, we can conclude that initiation is unaffected by oxydifficidin”, while the patterns observed on the gel for Ths (an established initiation inhibitor) and Oxy do not differ.

Response 7: Thank you, for your comment. We decided to re-design the experiment and add a control antibiotic, Edein A. As is known, Edeine А interferes with mRNA binding by overlapping with the codon location in the P-site and prevents the formation of the initiation complex, which is confirmed by the absence of ribosome stalling during toeprinting. Therefore, in this experiment, we do not visualize the release of BPy-Met.

The antibiotic thiostrepton binds to the L11 region and interferes with IF2 binding (initiation) and delivery of aa-tRNA to the ribosome's A-site by the EF-Tu factor (translocation). Since there is an excess of ribosomes over the template in this system, and IF2 inhibition is competitive, the 70S initiation complex is assembled, and the effect we would observe would be more likely to be due to the influence on elongation factors. As was shown, thiostrepton is able to shift the dynamic equilibrium of 70-S <-> 50-S + 30-S more towards dissociation (10.1111/j.1432-1033.1975.tb02324.x), though the effect of this antibiotic is primarily directed at translocation and the peptidyl transferase reaction (https://pmc.ncbi.nlm.nih.gov/articles/PMC3245911/). Since our template encodes only one amino acid, methionine, (AUG-UAA), elongation does not occur, and therefore we cannot observe the effect of these processes. For oxydifficidin, at any concentration, the release of the BPy-Met product is observed, which indicates the absence or weak effect on translation initiation.

 Comment 8: FP-based assays: as far as I could get from the Methods, pDualrep2 and pDualrep2.1 differ only by the resistance marker used (AmpR vs KanR). Why do the assays using these two plasmids (Figure 3) produce different signals (Red and Green pseudocolors) for the same control antibiotic Ery? How do the authors explain the same signal detected for Lev and Ery for pDualRep2.1?

Response 8: Thank you for your comment. In fact, pDualRep2 and pDualRep2.1 differ in more than just their resistance markers. While the reporter module—responsible for expressing fluorescent peptides that indicate antibiotic’s MOA—is identical in both systems, their enhanced sensitivity to antibiotics arises from distinct mutations present in the corresponding host strains.

The E. coli ΔtolC strain carrying pDualRep2, previously reported by our group, was derived from the E. coliJW5503 strain and lacks the tolC gene, which encodes the TolC protein—a crucial outer membrane channel in Gram-negative bacteria. TolC functions as an efflux pump for toxins, including antibiotics. Consequently, the ΔtolC pDualRep2 strain lacks this efflux mechanism and cannot expel harmful compounds, making it highly susceptible to any substance that can enter the cell. However, it remains resistant to molecules that are unable to penetrate the cell due to their size or other physicochemical properties.

To develop an even more sensitive reporter strain—an essential goal for detecting potential antibiotic pharmacophores—we utilized the E. coli SQ110ΔlptD strain, derived from E. coli BW25113. This strain lacks the lptD gene, which encodes an essential protein involved in the final stages of lipopolysaccharide assembly in the outer membrane. The 23-amino-acid deletion in lptD impairs its function, rendering the outer membrane more permeable to antibiotics.

We understand the confusion resulting from insufficiently detailed explanation of pDualRep2 and pDualRep2.1 constructions and expanded section 3.4.1. for clarification.

Giving the difference between pDualRep2 and pDualRep2.1, the accumulation of fluorescent proteins at sublethal concentrations of the tested antibiotics occurs over slightly different time periods. In the experiments described in the initial version of the manuscript, both reporter systems were tested in parallel, which resulted in the ambiguous outcomes you correctly noted. We have since repeated the experiment for pDualRep2.1 independently, optimizing the incubation time, and obtained consistent results for both control antibiotics across both reporter systems. Fig 3B were replaced.

Comment 9: In addition, the work is difficult to read; the Results section contains very technical parts more appropriate for the Methods section (e.g., lines 182-186 and the whole part 2.2). The work lacks a coherent narrative (e.g., the assay with fluorestent protein Katushka2S is first used in section 2.2, while it is explained only later in 2.4.1).

Response 9: Thank you for your comments regarding the structure of the manuscript; we have carefully considered them in the revised version. While some technical aspects are now described in detail in the Materials and Methods section, we believe that completely omitting Section 2.2 would not be appropriate. The isolation of an active compound from a complex biological mixture is not merely a technical procedure but a key part of the study’s conceptual framework. Moreover, it would be illogical to present the producer strain and then proceed directly to the characterization of the biologically active secondary metabolite without first describing how this compound was obtained in its purified form.

Comment 10: To sum up, the work does not present a clearly substantiated advancement regarding the mode of action of oxydifficidin beyond previously established findings. As such, in my opinion, the manuscript does not meet the novelty and mechanistic insight standards for publication in the International Journal of Molecular Sciences.

Response 10: We thank the reviewer for this assessment and the opportunity to clarify our contribution. As outlined in our responses above, our study provides additional mechanistic insight into oxydifficidin’s mode of action—specifically its effect on translation elongation rather than initiation—along with new evidence on its low cytotoxicity and the characteristics of its Bacillus producers. We believe these findings represent a meaningful extension of current knowledge on this compound.

Reviewer 2 Report

Comments and Suggestions for Authors

The text provides important evidence on oxydifficidin: it is a translation inhibitor with eukaryotic selectivity. This is a timely topic of interest for the Journal. It requires some adjustments to be optimal, but in my opinion, it is ready for publication after minor revisions.

Abstract

Lines 27–28: The word “antibiotic” appears twice in the same sentence.

Line 32: Please verify the sentence. Is there truly a gradient of increase? Or did you mean “induces/causes”?

Line 41: There is a grammatical error in “as a therapeutic activity”; I suggest “as a therapeutic agent.”

Introduction

Lines 49–54: The argument is good; re-evaluating overlooked antibiotics is key for this article and could be developed in greater depth.

Lines 70–73: I suggest briefly stating the methods and the hypothesis. Results and Discussion

Lines 83–88: The statement “identity of the strain is 96% or higher, which proves that it is the same strain” is risky. ≥96% identity does not imply “same strain”; better say “same species” or “closely related.”

Lines 93–95: Cite concrete values (e.g., ANI ≥98% with FZB24, if applicable).

Line 99: “Well-supported” with 81% support: in current phylogenomics, 81% is considered moderate. 

Table 10: The legend should indicate the tool, version, and parameters used to calculate identities.

Lines 125–127: Culture conditions for metabolite production and the use of reporter strains belong in Methods; they should not be in this section.

Recommendations

The Discussion should acknowledge limitations of prior work and contrast what was known vs. what is contributed here and what remains to be resolved; this will strengthen the manuscript.

The section on agricultural potential should be supported with more data on B. velezensis as a plant growth–promoting and biocontrol species.

Materials and Methods

Line 301: Is there a genomic accession number associated with the strain? If so, please add it.

Lines 324–330: Please add OD600 at harvest (or the target physiological phase) and indicate whether media were sterile and by which method.

Lines 343–361: Please include software/version/parameters used for assembly, coverage metrics, and the annotation tool.

Lines 374–376: Please include all phylogenetic analysis parameters (software/version, model or distance metric, number and type of support replicates, rooting criterion) and the taxonomic thresholds applied.

Lines 426–429: Suggestion: include percent identity values.

Author Response

We thank the reviewer for their constructive comments, which have helped us improve the manuscript. In response to these comments and our thoughts on the text, we have performed additional experiments, including the toe-printing and short-peptide synthesis assays, and evaluated oxydifficidin on our reporter system. The manuscript has been substantially revised accordingly. Our point-by-point responses are detailed below (in red).

Comment 1. The text provides important evidence on oxydifficidin: it is a translation inhibitor with eukaryotic selectivity. This is a timely topic of interest for the Journal. It requires some adjustments to be optimal, but in my opinion, it is ready for publication after minor revisions.

Response 1. Thank you for your positive feedback. We appreciate your recognition of the importance of our work on oxydifficidin. We are grateful for your constructive suggestions and carefully addressed all the minor revisions to improve the manuscript.

Comment 2. Abstract

Lines 27–28: The word “antibiotic” appears twice in the same sentence.

Response 2. Thank you for that comment. We rephrased the sentence to avoid tautology.

Comment 3. Line 32: Please verify the sentence. Is there truly a gradient of increase? Or did you mean “induces/causes”?

Response 3. We changed the wording to avoid misunderstanding.

Comment 4. Line 41: There is a grammatical error in “as a therapeutic activity”; I suggest “as a therapeutic agent.”

Response 4. We changed that per your suggestion.

Comment 5. Introduction

Lines 49–54: The argument is good; re-evaluating overlooked antibiotics is key for this article and could be developed in greater depth.

Response 5. We added a discussion about the importance of reviving old antibiotics.

Comment 6. Lines 70–73: I suggest briefly stating the methods and the hypothesis.

Response 6. The final paragraph of the Introduction has been revised and expanded to more thoroughly outline the scope and objectives of the study.

Comment 7. Results and Discussion

Lines 83–88: The statement “identity of the strain is 96% or higher, which proves that it is the same strain” is risky. ≥96% identity does not imply “same strain”; better say “same species” or “closely related.”

Response 7. Thank you for your comment. We rephrased that.

Comment 8. Lines 93–95: Cite concrete values (e.g., ANI ≥98% with FZB24, if applicable).

Response 8. We added this information.

Comment 9. Line 99: “Well-supported” with 81% support: in current phylogenomics, 81% is considered moderate. 

Response 19. Thank you for suggestion, we have changed “well” to “moderate”.

Comment 10. Table 10: The legend should indicate the tool, version, and parameters used to calculate identities.

Response 10. We have added this information in table 1.

Comment 11. Lines 125–127: Culture conditions for metabolite production and the use of reporter strains belong in Methods; they should not be in this section.

Response 11. We removed excessive experimental details that are already covered in the Methods section.

Comment 12. Recommendations

Response 12. The Discussion should acknowledge limitations of prior work and contrast what was known vs. what is contributed here and what remains to be resolved; this will strengthen the manuscript.

Thank you for sharing your comment. We have paid more attention to this aspect in the manuscript. We decided to include this in the introduction to emphasize the scope of our work.

Comment 13. The section on agricultural potential should be supported with more data on B. velezensis as a plant growth–promoting and biocontrol species.

Response 13. We have added this in the begging of results and discussion section.

Comment 14. Materials and Methods

Response 14. Line 301: Is there a genomic accession number associated with the strain? If so, please add it.

Genomic accession number CP199744.1 for EV17 is stated in table 2.

Comment 15. Lines 324–330: Please add OD600 at harvest (or the target physiological phase) and indicate whether media were sterile and by which method.

Response 15. We have added this information.

Comment 16. Lines 343–361: Please include software/version/parameters used for assembly, coverage metrics, and the annotation tool.

Response 16. All details were added in the text.

Comment 17. Lines 374–376: Please include all phylogenetic analysis parameters (software/version, model or distance metric, number and type of support replicates, rooting criterion) and the taxonomic thresholds applied.

Response 17. This information was added in the text.

Comment 18. Lines 426–429: Suggestion: include percent identity values.

Response 18. We have added these values.

Round 2

Reviewer 1 Report

Comments and Suggestions for Authors

In the updated version of the manuscript, the authors comprehensively addressed the major issues I raised previously in my initial review by the data from additional experiments or repeats of the existing ones with the addition of valid controls. It made the conclusions drawn more solid and better supported in the form they are presented in the current version. Even more importantly, the authors shifted the paradigm and overall scope of the paper from just the mechanism of oxydifficidin to the characterization of the producing strains and other aspects crucial for further drug development, such as the lack of cytotoxicity and inability of eukaryotic translation inhibition. Although I still believe that the reported insights into the drug action per se do not significantly push us closer to the functional understanding of the molecule’s MOA, in the current form, the paper serves as a valuable source of information for scientists interested in this overlooked family of polyketides in many aspects. In the current form, the paper can be considered for publishing in IJMS after the authors address the following remaining (mainly minor) issues:

  • Line 102: unclear why the amyE gene name is underlined
  • Line 109: “they belong” instead of “it belongs”?
  • Line 115: siderophore cannot be “confirmed in the genome”, only the presence of its BGC. Could you please outline it correctly.
  • Line 119: ISR stands for Induced Systemic Resistance; consider using the verb “triggering ISR” here
  • Line 132: moderately?
  • Fig 1: Please report the tree scale for the phylogenetic tree
  • Line 145: given that you apply only BLAST-based ANI here, I would call it just ANI, not ANIb
  • Line 160: reporting inhibition zone size makes little sense here, as you specify neither the test strain nor the amount of the medium applied
  • Line 161: I still insist that this assay with two fluorescent proteins should be explained here, where it is first used, not later, otherwise the reader is confused and does not understand the result
  • Line 173/ Line 178: The masses reported in the text for [M+Na]+ and [M-H]- reported in the text do not match the masses provided in Fig. S2B (583.2828 vs 583.2834 and 559.2708 vs 559.2698). Please correct
  • Fig 2B: for oxydifficidin, please provide the following in the figure: theoretical monoisotopic mass, your experimental masses in + and – ion modes, and the corresponding errors in ppm.
  • Fig 2B: Please consistently label the observed ions peaks in [ ] and provide the charge for each ion
  • Fig 2B: The y and x axis labels are too small and unreadable
  • Fig 2A: The use of a very smooth gradient for the BGC is not the best choice, as it precludes unambiguous assignment of gene names. Please label the genes above every arrow in the BGC
  • Line 198: TolC serves multiple pumps
  • Line 203-204: a fragment of a sentence is missing, apparently
  • Fig 3: It would be great to see the two controls in the same left-right orientation on both panels; the name of the lptD gene should be italicized here and in several other places in the text.
  • Fig 4B: E. coli
  • Fig 4A/S5: could you please provide the gels for the three replicates of the toeprinting assay for the templates ermCL and ermDL you mention in your point-by-point response and methods section as raw images supplementing the manuscript resubmission
  • Line 310: if using classification of RPs from Ban et al. 2014, please stick to it consistently (L7/L12 => bL12)
  • Line 593: “6% polyacrylamide gel containing 19% acrylamide” is confusing to me.
  • Line 681: potency is a characteristic of a drug, not a mechanism
  • Fig S1: red labels are poorly visible; labels contain some non-English (Russian?) characters.
  • Figs S3 and S4: positive and negative modes are apparently switched between these figure legends.
  • Table S1: trehalose is misspelled

Author Response

We are grateful to the reviewer for their thoughtful comments and positive feedback on our revised manuscript. The clarity and rigor of our work have been further strengthened by their insightful observations, which have been invaluable. Our detailed responses are provided below after carefully addressing all the points raised.

Comment 1. Line 102: unclear why the amyE gene name is underlined

Response 1. Thank you for that comment. We have resolved that issue.

Comment 2. Line 109: “they belong” instead of “it belongs”?

Response 2. Thank you for highlighting this point. We have amended the text to rectify the issue.

Comment 3. Line 115: siderophore cannot be “confirmed in the genome”, only the presence of its BGC. Could you please outline it correctly.

Response 3. Thank you. We have amended the text to rectify the issue.

Comment 4. Line 119: ISR stands for Induced Systemic Resistance; consider using the verb “triggering ISR” here

Response 4. Thank you for this constructive suggestion. We have implemented the change.

Comment 5. Line 132: moderately?

Response 5. Thank you for that comment. We have resolved that issue.

Comment 6. Fig 1: Please report the tree scale for the phylogenetic tree

Response 6. Thank you! The picture has been modified.

Comment 7. Line 145: given that you apply only BLAST-based ANI here, I would call it just ANI, not ANIb.

Response 7. Thank you. The concern has been thoroughly attended to in the current version.

Comment 8. Line 160: reporting inhibition zone size makes little sense here, as you specify neither the test strain nor the amount of the medium applied

Response 8. We thank the reviewer for this comment. We resolved this issue.

Comment 9. Line 161: I still insist that this assay with two fluorescent proteins should be explained here, where it is first used, not later, otherwise the reader is confused and does not understand the result

Response 9. Our thanks to the reviewer for highlighting this point. We have transferred text here from version 2.2, including the reporter's description.

Comment 10. Line 173/ Line 178: The masses reported in the text for [M+Na]+ and [M-H]- reported in the text do not match the masses provided in Fig. S2B (583.2828 vs 583.2834 and 559.2708 vs 559.2698). Please correct

Response 10. Thank you for pointing this out. The masses were measured multiple times, and the value reported in the text came from an earlier measurement. We have now corrected the masses reported in the text to match those shown in the Figure S2B.

Comment 11. Fig 2B: for oxydifficidin, please provide the following in the figure: theoretical monoisotopic mass, your experimental masses in + and – ion modes, and the corresponding errors in ppm.

Fig 2B: Please consistently label the observed ions peaks in [ ] and provide the charge for each ion

Fig 2B: The y and x axis labels are too small and unreadable

Fig 2A: The use of a very smooth gradient for the BGC is not the best choice, as it precludes unambiguous assignment of gene names. Please label the genes above every arrow in the BGC

Response 11. Thank you for your input. We incorporated all of your suggestions to improve Figure 2. We labelled the genes above arrows in the BGC. In Figure 2B we added the theoretical monoisotopic mass together with our experimental mass measured in positive-ion mode and the corresponding ppm error. We chose not to include the negative-ion data because the absolute peak intensity was too low to provide reliable mass accuracy, leading to unusually high ppm deviations. This is a well-known effect associated with low-intensity ions, which yield less stable centroid determination, particularly in negative-ion mode where some analytes ionize less efficiently.

Comment 12. Line 198: TolC serves multiple pumps

Response 12. Thank you for your careful reading; we have corrected this issue and used the plural form of the word pump in this sentence.

Comment 13. Line 203-204: a fragment of a sentence is missing, apparently

Response 13. Thank you. We have corrected this issue.

Comment 14. Fig 3: It would be great to see the two controls in the same left-right orientation on both panels; the name of the lptD gene should be italicized here and in several other places in the text.

Response 14. Thank you for this suggestion. We have adjusted the figures accordingly and corrected the italicization of the lptD gene throughout the text

Comment 15. Fig 4B: E. сoli

Response 15. Thank you for pointing that out, we changed this label in Figure 4B.

Comment 16. Fig 4A/S5: could you please provide the gels for the three replicates of the toeprinting assay for the templates ermCL and ermDL you mention in your point-by-point response and methods section as raw images supplementing the manuscript resubmission

Response 16. Three replicates of the toeprinting assay for the ermCL and ermDL templates (both fluorescent and radioactive) have been added to the Non-published Material section

Comment 17. Line 310: if using classification of RPs from Ban et al. 2014, please stick to it consistently (L7/L12 => bL12)

Response 17. Thank you. We have revised the text, updating L7/L12 to bL12, and adjusted the labeling in Figure S7 accordingly.

Comment 18. Line 593: “6% polyacrylamide gel containing 19% acrylamide” is confusing to me.

Response 18. Thank you for your comment. This sentence referred to the ratio of components, and we have rewritten it in the Materials and Methods section to avoid any misunderstandings.

Comment 19. Line 681: potency is a characteristic of a drug, not a mechanism

Response 19. Thank you for pointing this out; that was indeed an unfortunate choice of wording. We have revised the sentence to ensure it is accurate.

Comment 20. Fig S1: red labels are poorly visible; labels contain some non-English (Russian?) characters.

Response 20. Thank you. Figure S1 labels were adjusted to improve readability.

Comment 21. Figs S3 and S4: positive and negative modes are apparently switched between these figure legends.

Response 21. Thank you. We have updated the figure legends accordingly.

Comment 22. Table S1: trehalose is misspelled

Response 22. Thank you. We have corrected this typo and reviewed the text for any others.
